# Range-wide phylogenomics of the Great Horned Owl (*Bubo virginianus*) reveals deep north-south divergence in northern Peru

Emily N. Ostrow, Lucas H. DeCicco and Robert G. Moyle

Biodiversity Institute and Department of Ecology and Evolutionary Biology, University of Kansas, Lawrence, KS, United States of America

## ABSTRACT

The Great Horned Owl (*Bubo virginianus*) inhabits myriad habitats throughout the Americas and shows complex patterns of individual and geographic morphological variation. The owl family Strigidae is known to follow ecogeographic rules, such as Gloger's rule. Although untested at the species level, these ecogeographic rules may affect *B. virginianus* plumage coloration and body size. Previous studies have indicated that, despite this species' morphological variability, little genetic differentiation exists across parts of their range. This study uses reduced representation genome-wide nuclear and complete mitochondrial DNA sequence data to assess range-wide relationships among *B. virginianus* populations and the disputed species status of *B. v. magellanicus* (Magellanic or Lesser Horned Owl) of the central and southern Andes. We found shallow phylogenetic relationships generally structured latitudinally to the north of the central Andes, and a deep divergence between a southern and northern clade close to the Marañón Valley in the central Andes, a common biogeographic barrier. We identify evidence of gene flow between *B. v. magellanicus* and other subspecies based on mitonuclear discordance and F-branch statistics. Overall differences in morphology, plumage coloration, voice, and genomic divergence support species status for *B. v. magellanicus*.

Corresponding author
Emily N. Ostrow,
emily.ostrow@ku.edu

## INTRODUCTION

The Great Horned Owl (*Bubo virginianus*) is a common and widespread species throughout the Americas (*Artuso et al., 2013*), occurring in habitats with diverse environmental conditions, such as desert, coastal rainforest, and high-elevation montane forest. The species occurs from the southern tip of Argentina to the northern edge of boreal forest in North America (*Artuso et al., 2013*). *Bubo virginianus* also exhibit complex inter- and intra-population variation in body size and plumage coloration (Fig. 1) including sex-based and individual variation (*Mattison & Witt, 2021*; *Pyle, 1997* p 75–78). Fifteen *B. virginianus* subspecies are recognized based on geographic variation of plumage color and pattern (*Gill, Donsker & Rasmussen, 2022*). The South American subspecies *B. v. magellanicus* was

recently recognized as a full species by *Clements et al. (2022)*, but opinions are divided on this decision (see South American Checklist Committee proposal 328). We refer to *B. v. magellanicus* as a subspecies here due to the disputed species status. Phenotypic variation manifests in differences in size, in overall lightness and redness of plumage coloration, as well as the distribution and saturation of melanin-based pigmentation (*Weidensaul, 2015*). Extremes in phenotype are represented by the pale northern boreal subspecies, *B. v. subarcticus*, and the comparatively small and dark Pacific Northwest subspecies *B. v. saturatus*. Due to the continental distribution of the species and the lack of stark biogeographic boundaries among described taxa, it is likely that most described subspecies show clinal variation (*Dickerman, 1993*). These factors complicate morphology-based taxonomic determination in this species. The complexity of individual and sex-based differences, in conjunction with the broader but often poorly differentiated geographic variation makes *B. virginianus* one of the more complicated and poorly understood avian species that spans North and South America.

Another factor that further complicates our current understanding of population-level differentiation within *B. virginianus* is short distance migration and long-distance dispersal in some populations (*Dickerman, Mcnew & Witt, 2013*). Literature on the movement of *B. virginianus* during non-breeding season is conflicting. *Baumgartner (1939)* observed owls roosting in their nesting sites year-round in Lawrence, Kansas; however, short-distance migration is evident in some northern populations of *B. virginianus*, likely due to harsher winters (*Dickerman, Mcnew & Witt, 2013*; *Holt, 1996*; *Houston & Francis, 1995*; *Houston, 1999*). Short distance migration is most common in northern populations and has only been demonstrated in *B. v. subarcticus,* although *Dickerman (1993)* has suggested that it may be common in other northern populations such as *B. v. lagophonus* (*Houston & Francis, 1995*; *Houston, 1999*). This pattern of migration however has mainly been observed through bird banding efforts, so there is little information on whether this movement happens yearly or based on other factors such as food availability (*Houston & Francis, 1995*; *Houston, 1999*).

Although phenotypic variation in this species is substantial and confusing, vocal variation appears much more constrained. For example, little song variation exists among North American populations (*López-Lanús, 2015*). Previous studies on vocal differentiation in *B. virginianus* have indicated that the differentiation of *B. v. magellanicus* songs might warrant species status of this taxon (*López-Lanús, 2015*; *König, Heidrich & Wink, 1996*). *López-Lanús (2015)* conducted a thorough analysis of *B. virginianus* songs throughout its range and identified *B. v. magellanicus* and *B. v. nigrescens* as a putative species based on vocal analyses. Phenotypic studies have not suggested that *B. v. nigrescens* should be considered a full species, but no genetic work has focused on South American *B. virginianus* (*Traylor, 1958*). Songs of non-passerines such as owls are considered innate and therefore the relatively simple vocal variation in this species may be phylogenetically informative (*Isler, Isler & Brumfield, 2005*; *Sangster et al., 2013*).

In comparison with the extensive literature on morphological variation and some characterization of vocal variation in *B. virginianus*, relatively little is known about the genetic variation within the species. A family-level phylogeny by *Wink et al. (2009)* using one nuclear and one mitochondrial gene determined that *B. virginianus* was sister to *B.*

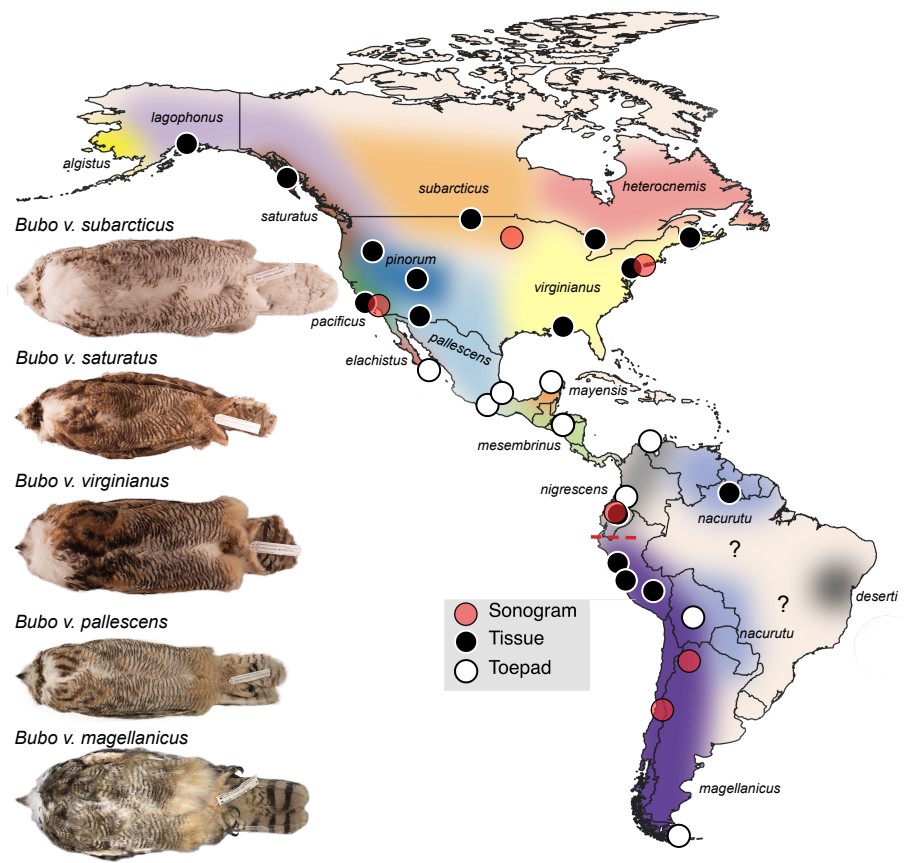

**Figure 1** **Sampling map of specimens and recordings used in this study.** Black dots represent fresh tissue samples and white dots represent toepad samples. Red dots are locations where the songs were recorded for the sonograms in Fig. 4. Pictured are example specimens of five subspecies of *Bubo virginianus* including: *B. v. subarcticus* (KU 81018 from USA: Minnesota), *B. v. saturatus* (KU 94293 from USA: Oregon), *B. v. virginianus* (KU 135711 from USA: Kansas), *B. v. pallescens* (KU 135933 from USA: Kansas), and *B. v. magellanicus* (LSUMZ 68779 from Peru: Dept. Junin). These individuals show the range in size, lightness, and redness in the species. The individuals photographed are not necessarily included as genetic samples. Approximate subspecies distributions are represented by various colors. Areas with question marks have unknown population affinities. Subspecies distributions were outlined based on *Clements et al. (2022)* distribution descriptions. We have indicated uncertainty by not displaying hard outlines separating subspecies. The dashed red line shows the approximate location of the Marañón valley.

*scandiacus* (Snowy Owl). Information on patterns of intra-specific genetic variation is scant but suggests low genetic variation and no genetic structure across the southwestern United States (*Dickerman, Mcnew & Witt, 2013*). Plumage, vocal, and limited genetic data indicate that the southern South American *B. v. magellanicus*, occurring in the central and southern Andes, is divergent enough to be considered a distinct species (*e.g., Clements et al., 2022*; *Gill, Donsker & Rasmussen, 2022*; *López-Lanús, 2015*; *König, Heidrich & Wink, 1996*). The International Ornithological Congress recognizes *B. v. magellanicus* as a species (*B. magellanicus; Gill, Donsker & Rasmussen, 2022*); however, species status was disputed by the South American Classification Committee (proposal 328) because the only genetic

study was based on two individuals, one *B. v. magellanicus* and one non-*magellanicus B. virginianus* using sequences of one gene only (*König, Heidrich & Wink, 1996*). Genomic-scale data are needed to understand the range-wide genetic structure of *B. virginianus* populations.

Here, we present the first genome-wide estimate of genetic differentiation in Great Horned Owls across their entire distribution. Specifically, we (1) characterize range-wide genetic variation and differentiation within the species by sampling thirteen of the sixteen recognized subspecies (subspecies not sampled are *heterocnemis, algistus,* and *deserti*), and (2) evaluate the putative species status of the populations of southwestern South America (subspecies *magellanicus*). We used reduced-representation nuclear genomic and mitochondrial sequence data to reconstruct a phylogeny and address the question of species status in the broader context of variation across the range of the *B. virginianus* complex.

## MATERIALS & METHODS

### Taxon sampling

Twenty-seven samples of *Bubo virginianus* were assembled from across the range of the species. Specimen-vouchered tissue and toepad samples were loaned from sixteen museum collections (Table 1). This sampling includes thirteen of the sixteen recognized subspecies (*Clements et al., 2022*); subspecies assignment was based on collection locality and described distributions of subspecies in the literature (Fig. 1). When possible, sampling prioritized specimens taken during the regional nesting season (Table 1). Sequence data for three outgroup species (*B. cinerascens*, *B. nipalensis*, and *B. scandiacus*) were downloaded from the Sequence Read Archive (*Salter et al., 2020*; Table 1).

### DNA extraction and sequencing

Two DNA extraction methods were used, depending on tissue type. DNA was extracted from toepad samples using a Promega Maxwell RSC automated instrument and the Maxwell RSC Tissue DNA Kit. This instrument minimizes potential for contamination, which is important for degraded DNA samples. Fresh tissue samples were extracted using a manual bead-extraction protocol (https://github.com/phyletica/lab-protocols/blob/master/extraction-spri.md), based on *Rohland & Reich (2012)*, with elution of DNA in 1 X TE buffer.

After extraction, we sonicated all fresh tissue samples using a Covaris M220 sonicator (50 W peak incident power, 20% duty factor, and 200 cycles per burst for 65 s). DNA from toepad samples was already relatively degraded, and therefore did not need to be sonicated (*McCormack, Tsai & Faircloth, 2016*). We then enriched for ultraconserved elements (UCEs) following a standard protocol using the Mycroarray MYbaits kit for Tetrapods UCE 5K v1 (*Faircloth et al., 2012*). During pooling, on average, each fresh tissue sample received 2.2 million reads per sample and each toepad received 5 million reads per sample. All UCE libraries were sequenced using Illumina paired end 150 bp sequencing on a high output run of a NextSeq550 machine at the KU Genome Sequencing Core. Raw sequence files were deposited in the Sequence Read Archive (Table 1).

**Table 1  Sampling table of the *Bubo virginianus* specimens used in this study.**

| Species name | NCBI number | Specimen number | Institution | Month | Tissue type | Locality |
|---|---|---|---|---|---|---|
| *B. v. virginianus* | SRR21537326 | 159674 | ROM | May | Tissue | Canada, Ontario |
| *B. v. virginianus* | SRR21537383 | 13382 | NBMB | Feb | Tissue | Canada, New Brunswick |
| *B. v. lagophonus* | SRR21537325 | 24809 | UAM | May | Tissue | United States, Alaska |
| *B. v. saturatus* | SRR21537314 | 6917 | UAM | Mar | Tissue | United States, Alaska |
| *B. v. pacificus* | SRR21537307 | 115875 | LACM | Sep | Tissue | United States, California |
| *B. v. subarcticus* | SRR21537306 | 88562 | KUNHM | Jun | Tissue | United States, North Dakota |
| *B. v. pinorum* | SRR21537305 | 23281 | MSB | Jul | Tissue | United States, Utah |
| *B. v. pinorum* | SRR21537304 | 74120 | UWBM | Apr | Tissue | United States, Oregon |
| *B. v. pinorum* | SRR21537303 | 45709 | MSB | Jun | Tissue | United States, Arizona |
| *B. v. virginianus* | SRR21537302 | 396867 | FMNH | Mar | Tissue | United States, Florida |
| *B. v. virginianus* | SRR21537301 | 203198 | ANSP | Jan | Tissue | United States, Pennsylvania |
| *B. v. pallescens* | SRR21537324 | 63660 | CAS | Jan | Toepad | Mexico, Guerrero |
| *B. v. elachistus* | SRR21537323 | 21432 | DMNS | Apr | Toepad | Mexico, Baja California Sur |
| *B. v. mayensis* | SRR21537322 | 187122 | FMNH | Sep | Toepad | Mexico, Veracruz |
| *B. v. mayensis* | SRR21537321 | 286522 | MCZ | Oct | Toepad | Mexico, Yucatan |
| *B. v. mesembrinus* | SRR21537320 | P135334 | CM | Aug | Toepad | Honduras, Francisco Moraz |
| *B. v. mesembrinus* | SRR21537319 | P135270 | CM | Aug | Toepad | Honduras, Francisco Moraz |
| *B. v. nigrescens* | SRR21537318 | 102981 | FMNH | Sep | Toepad | Colombia, Cauca |
| *B. v. nigrescens* | SRR21537317 | 368745 | USNM | May | Toepad | Colombia, La Guajira |
| *B. v. nigrescens* | SRR21537316 | 30041 | LSUMZ | Aug | Tissue | Ecuador, Napo |
| *B. v. nacurutu* | SRR21537315 | 90879 | KUNHM | Apr | Tissue | Guyana |
| *B. v. magellanicus* | SRR21537313 | 214109 | UMMZ | Jun | Toepad | Bolivia, Cochabamba |
| *B. v. magellanicus* | SRR21537312 | 36086 | MSB | May | Tissue | Peru, Ancash |
| *B. v. magellanicus* | SRR21537311 | 61413 | LSUMZ | Mar | Tissue | Peru, Cusco |
| *B. v. magellanicus* | SRR21537310 | 35949 | MSB | Mar | Tissue | Peru, Lima |
| *B. v. magellanicus* | SRR21537309 | 35837 | MSB | Aug | Tissue | Peru, Lima |
| *B. v. magellanicus* | SRR21537308 | 120621 | FMNH | Dec | Toepad | Argentina, Tierra del Fuego |
| *B. cinerascens* | SRX7052782 | 15360 | KUNHM | Oct | Tissue | Ghana, Upper West Region |
| *B. nipalensis* | SRX7052765 | 189733 | FMNH | Jan | Toepad | India, Madhya Pradesh |
| *B. scandiacus* | SRX7052783 | 27634 | KUNHM | Mar | Tissue | United States, Kansas |

**Notes.**

Subspecies are determined by location according to the *Clements et al. (2022)*. Institution acronyms correspond with the following: Academy of Natural Sciences of Drexel University (ANSP), California Academy of Sciences (CAS), Carnegie Museum of Natural History (CM), Denver Museum of Nature & Science (DMNS), Field Museum of Natural History (FMNH), University of Kansas Biodiversity Institute (KUNHM), Natural History Museum of Los Angeles County (LACM), Louisiana State University Museum of Natural Science (LSUMZ), Museum of Comparative Zoology (MCZ), Museum of Southwestern Biology (MSB), Royal Ontario Museum (ROM), University of Alaska Museum (UAM), University of Michigan Museum of Zoology (UMMZ), National Museum of Natural History (USNM), University of Washington Burke Museum (UWBM).

## UCE sequence data assembly

All UCE data were processed using the standard Phyluce pipeline with the mapping and correction workflows (*Faircloth et al., 2012*). Briefly, we cleaned raw reads using Illumiprocessor (version 2.0.9), a wrapper for Trimmomatic (version 0.39, *Bolger, Lohse & Usadel, 2014*) and assembled clean reads using the trinity (version 2.8.5, *Grabherr et al., 2011*) assembler in Phyluce (version 1.6.8, *Faircloth et al., 2012*). We then mapped reads onto the resulting contigs using the Phyluce mapping workflow. Bases were then removed

if they had a Phred score below 20, a depth below five, or greater than two alleles in called genotypes. These new corrected contigs were then matched with the tetrapods 5K probeset available at ultraconserved.org. These contigs were aligned using MAFFT (version 7.455, *Katoh & Standley, 2013*), and internally trimmed using gblocks (version 0.91B, *Castresana, 2000*). We then removed loci that were present in <75% of individuals. These data were split into two datasets, one from fresh tissues only and one with all samples. For the data that included all samples, we wrote a custom script that replaced phylogenetically uninformative sites in the alignment (https://github.com/emilyostrow/ReplaceUninformativeSites) due to potential data degradation in toepad samples. In addition to missing data, toepad samples tended to have erroneous bases sequenced near cut sites. This script replaced any base that is represented only once at a particular site with an N to address branch length concerns caused by degraded DNA.

After generating UCEs, we called single nucleotide polymorphisms (SNPs) using the consensus UCE sequences as a reference with the GATK pipeline. We used Geneious Prime (version 2023.0.4, https://www.geneious.com) to generate consensus sequences on all UCEs found in at least 75% of the individuals using a 50% strict cutoff for each site. To call SNPs, we first trimmed raw read files using AdapterRemoval (version 2.3.2, *Schubert, Lindgreen & Orlando, 2016*). We then built an index of the UCE consensus sequences in Bowtie2 (version 2.3.5.1, *Langmead & Salzberg, 2012*) and aligned reads to the reference using the very-sensitive-local algorithm. After aligning the reads, added read group information and marked duplicates in picard (version 2.20.3, http://broadinstitute.github.io/picard) and indexed the files using SAMtools (version 1.9 *Li et al., 2009*) We then used GATK (version 4.2.6.1, *McKenna et al., 2010*) to call haplotypes, genotype sequences, select SNPs, and filter SNPs. We hard filtered SNPs for quality by depth (QD) < 2.0, strand odds ratio (SOR) > 4.0, Fisher strand (FS) > 60.0, and RMS mapping quality (MQ) < 40.0. We then filtered the SNPs again using SNPfiltR (version 1.0.0, *DeRaad, 2022*) in R (version 4.1.1, *R Core Team, 2021*) for 90% or greater data completeness by SNP, phred score of 30 or greater and only included biallelic SNPs.

## Mitochondrial genome assembly

As a byproduct of UCE sequencing, the mitochondrial genome is often also sequenced (*Do Amaral et al., 2015*). To obtain mitochondrial genomes, we used the program MITObim following the two-step procedure outlined in the manual (version 1.9.1, *Hahn, Bachmann & Chevreux, 2013*). We used cleaned reads from the Illumiprocessor step in Phyluce as input data and a *B. scandiacus* sample (NC_038220.1) from NCBI as a reference mitochondrial genome for alignment. Briefly, we interleaved reads one and two from the cleaned Illumiprocessor data. We created an initial read map using MIRA (version 4.0.2, *Chevreux, Wetter & Suhai, 1999*), then completed iterative mapping using MITObim. MITObim ran up to ten iterations or until it reached a stationary read number.

All individual mitochondrial genome alignments were then aligned to the reference *B. scandiacus* sample using MAFFT in Geneious (version 8.1.9, *Kearse et al., 2012*) to create multi-sample alignments. Gaps in alignment created by a single individual were removed by hand. Non-genic regions were removed from the multi-sample alignment because more

variable areas such as the control region did not align well and may have led to inaccurate results.

## Phylogenetics

Phylogenetic analyses were completed with the UCE data using maximum-likelihood, quartet-based, and neighbor-joining approaches. For the maximum likelihood method, we used the program SWSC-EN (*Tagliacollo & Lanfear, 2018*) to split UCE data into conserved core and more variable flanking regions for each UCE. These regions were then tested as potential partitions in PartitionFinder2 using a rclusterf search scheme (version 2.1.1, *Lanfear et al., 2017*). We used the best partitioning scheme according to the AICc score with a GTR+G model in RAxML-NG (version 1.1.0, *Kozlov et al., 2019*) using both the only-tissues dataset and the full sampling with no uninformative sites. We completed only 200 bootstrap replicates of the tissues-only matrix due to bootstrap convergence and 1,000 bootstrap replicates of the full sampling matrix. Our quartet-based approach used SVDquartets in PAUP* (version 4.0a, *Chifman & Kubatko, 2015*) on default settings with 1,000 bootstrap replicates. We generated a neighbor-joining tree using the dataset with all individuals and no uninformative sites in Geneious Prime. We completed 1,000 bootstraps and collapsed nodes with a support threshold below 50%.

Our mitochondrial phylogenetic analyses included both maximum likelihood and Bayesian approaches. We used PartitionFinder2 to test models for both tree-building approaches. We split all genes by codon and tested all models of evolution using a greedy search scheme. Models and partitions were evaluated using AICc scores. Our maximum likelihood analysis was completed with RAxML-NG, using the partitions from PartitionFinder2 and a GTR+G model with 200 bootstrap replicates, after which the tree converged. Our Bayesian analysis was completed using the program MrBayes (version 3.2.7a, *Ronquist & Huelsenbeck, 2003*). We used the optimal partitions and models from PartitionFinder2. Our analysis used four chains with 10 million generations, using a burn-in fraction of 25%, sampling every 1,000 generations. We checked to ensure convergence for the Bayesian tree using potential scale reduction factor and effective sample size estimates.

## Gene flow

We tested for gene flow between populations using Dsuite (version 0.4 r42, *Malinsky, Matschiner & Svardal, 2021*). Specifically, we calculated the D-statistics and f4-ratios using the Dtrios algorithm using the vcf file and full sampling UCE tree. We then used the Fbranch algorithm to calculate the F-branch statistic for each individual and ancestor comparison. We also calculated *p*-values associated with the F-branch statistics and corrected for multiple testing using the False Discovery Rate Benjamini-Hochberg procedure (*Benjamini & Hochberg, 1995*). We visualized these results and support values using dtools.

## Audio recording sonograms

We downloaded six audio recordings of *B. virginianus* songs with minimal background noise from Xeno-Canto (https://xeno-canto.org): XC428421 from Minnesota, USA, XC511167 from New York, USA, XC548133 from California, USA, XC76397 from Napo, Ecuador, XC212734 from Salta, Argentina, and XC494437 from Santiago, Chile. Audio

recordings were chosen based on audio quality and geographic breadth. We imported these songs into Adobe Audition (Build 14.1.0.43) and iteratively used the denoise tool to reduce background noise in the sonograms until the song was clearly differentiated from the background noise. We then exported audio files from Adobe Audition and imported them into Raven Lite (version 2.0.1, *K. Lisa Yang Center, 2023*) where we visualized each song as a sonogram, adjusted contrast as needed for clarity, and saved each song as an image file.

## RESULTS

### UCE and mitochondrial data

We generated a mean of 4248 UCE loci per individual. Fresh tissue samples yielded a mean of 4318 UCE loci per individual, whereas toepad samples yielded a mean of 4127 loci per individual. The 75% matrix (*i.e.,* loci present in ≥22 of the 30 individuals) included 4299 UCE loci, with a 1.26 Mbp overall alignment length and an average UCE length of 796 bp for fresh tissues and 299 bp for the toepad samples. Our unfiltered SNP dataset included 7584 SNPs. After filtering for SNP completeness, phred score, and biallelic sites, our final dataset included 6556 SNPs.

The mitochondrial baiting method was successful for all individuals in this study. MITObim contigs had a mean read depth of 101X and a mean contig length of 18,967 bp. Fresh tissue samples had a mean read depth of 95X, whereas toepad samples had a mean read depth of 111X, presumably due to increased sequencing effort dedicated to toepad samples. After trimming to genic regions only, we generated an 11,347 bp alignment, with 1.09% missing data. One individual in particular, *B. v. virginianus* (FMNH 396867), had 23.05% missing data and disproportionately contributed to the overall missing data in the mitogenome matrix. The effects of these missing data and degradation, contamination, or sequencing error at the edges of gaps are apparent in the longer branch in the mitochondrial tree.

### Phylogenetics

Maximum-likelihood, quartet-based, and neighbor-joining methods using the nuclear data produced trees that were congruent at most moderate to highly supported nodes (bootstrap support above 90%); the trees mainly differed in topology at nodes with low support. The one topological difference with bootstrap support in the quartet-based tree (Fig. S1) was the placement of *B. v. nigrescens* (FMNH 102981), which was sister to *B. v. nacurutu* (KU 90879) + *B. v. nigrescens* (USNM 368745) in the SVDquartets tree whereas in all other full sampling trees, *B. v. nigrescens* (FMNH 102981) was sister to *B. v. nigrescens* (LSUMZ 30041). The one topological change between the maximum-likelihood tree and neighbor-joining tree was the placement of *B. v. nacurutu* (KU 90879) + *B. v. nigrescens* (USNM 368745), which was sister to *B. v. nigrescens* (FMNH 102981) + *B. v. nigrescens* (LSUMZ 30041) in the neighbor-joining tree (Fig. S2) but is within the mostly Central American clade in the maximum likelihood tree. Here, we present the maximum-likelihood tree (Fig. 2). The quartet-based SVDquartets tree had few nodes with high support (Fig. S1). However, both RAxML and SVDquartets trees showed strong support for a *B. v.*

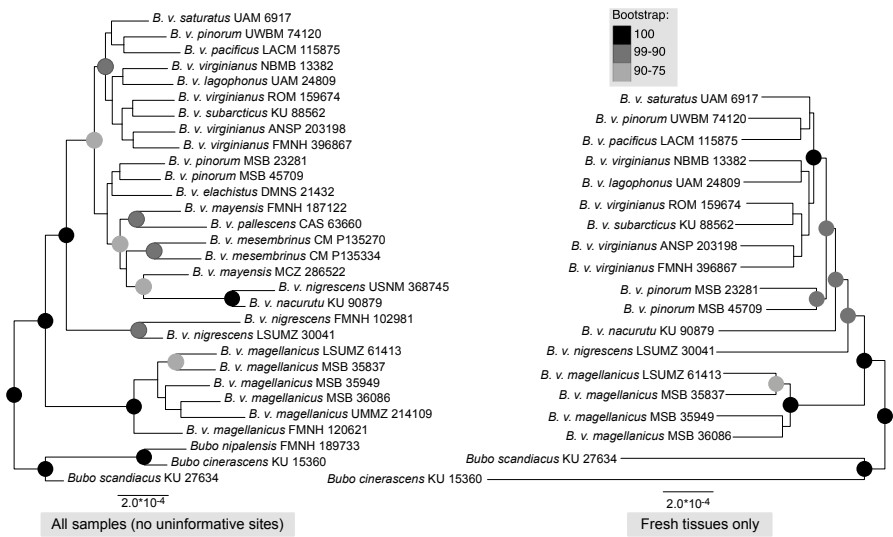

**Figure 2   Maximum-likelihood phylogenetic trees using UCE data.** Support are values shown in the shaded circles on nodes. All nodes without support values are not highly supported (<75 bootstrap support). The left tree includes all specimens sampled in this research including a combination of both toepad- and tissue-based genetic samples, the right tree includes only samples that were extracted from fresh tissue samples.

*magellanicus* clade sister to all other samples (hereafter referred to as the northern clade). Within the northern clade, lineages branching from earlier nodes belonged to more southern taxa (*B. v. nacurutu* and *B. v. nigrescens*) with less geographically concordant branching order in other northern subspecies.

Maximum-likelihood and Bayesian analyses using the mitochondrial dataset generally supported the same topology (Fig. 3, Fig. S3). Two discrepancies separated the trees, both at extremely shallow nodes: *B. v. mayensis* (FMNH 187122) and *B. v. pinorum* (UWBM 74120) switched positions, and *B. v. elachistus* (DMNS 21432) and *B. v. pacificus* (LACM 115875) switched positions. Support values from both analyses are mapped onto the RAxML tree at concordant nodes (Fig. 3).

The trees built using UCE and mitochondrial data were not entirely concordant. These inconsistencies might be due to the different inheritance of nuclear and mitochondrial DNA, gene flow, taxon sampling differences in the UCE trees, or potentially greater errors in toepad sequencing due to their degraded DNA. We attempted to address potential data errors by using the Phyluce correction pipeline and only using Phylogenetically informative sites to reduce spurious branch lengths. All trees shared a general pattern of southern lineages branching from earlier nodes, but the topologies contained strongly supported differences. The relationships of many of the taxa in southern Mexico, Central America, and northern South America are different in all reported trees. One major difference in the topologies is the relative placement of *B. v. nacurutu* (KU 90879) and *B. v. nigrescens* (LSUMZ 30041). In the fresh tissues only UCE tree, they are sequential sister taxa to the remainder of the northern clade, whereas in the all samples UCE tree, *B.*
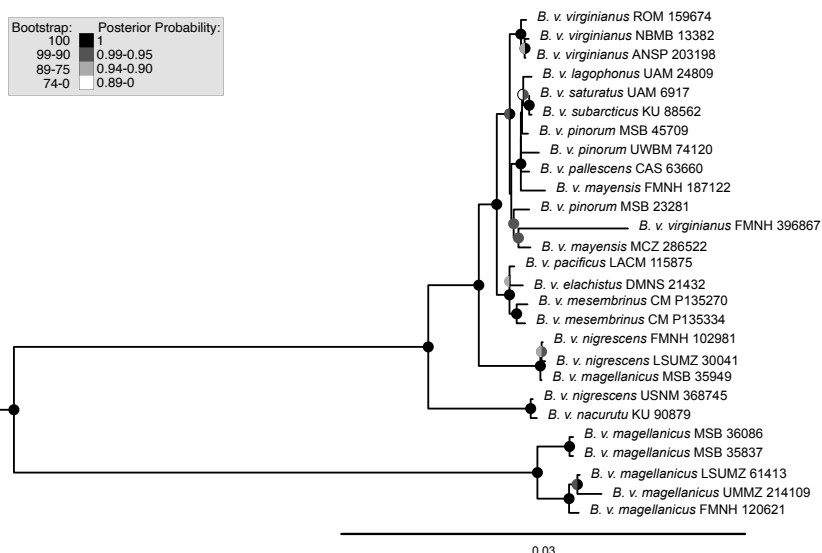

**Figure 3** **Maximum-likelihood phylogenetic tree using mitochondrial genome data.** Support values are shown in the shaded circles on nodes. Maximum-likelihood bootstrap support values are shown in the left half of the circles and Bayesian posterior probabilities are in the right half of the circles. Nodes without a circle have low support in both analyses.

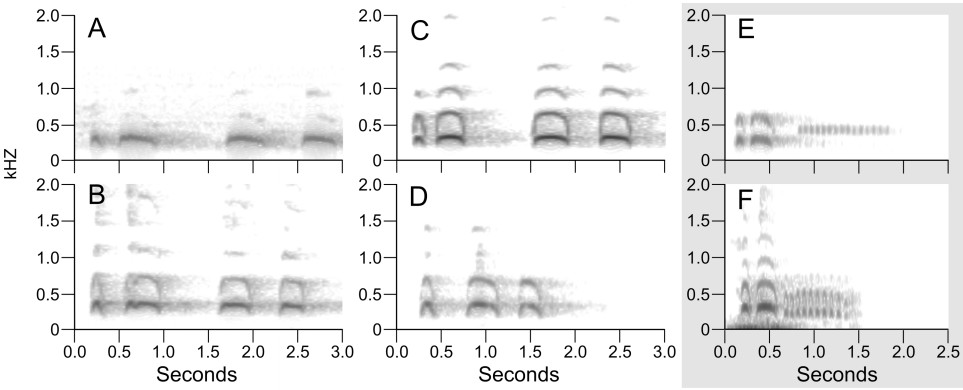

**Figure 4** **Example sonograms of audio recording of songs of northern *B. virginianus* (A–D) and *B. v. magellanicus* (E–F).** Sonograms are ordered north to south: (A) XC428421 from Minnesota, USA; (B) XC511167 from New York, USA; (C) XC548133 from California, USA; (D) XC76397 from Napo, Ecuador; (E) XC212734 from Salta, Argentina; (F) XC494437 from Santiago, Chile. Examples of *Bubo v. magellanicus* (E and F) are outlined in gray and characterized by a shorter song and a trill at the end of its song. Songs were downloaded from Xeno-Canto and locations for the songs are shown in Fig. 1.

*v. nacurutu* (KU 90879) is found within the larger North American clade (Fig. 2). In the mitochondrial tree, *B. v. nigrescens* (LSUMZ 30041) and *B. v. nacurutu* (KU 90879) switch branching order as compared to the UCE trees (Fig. 3). Similar to the UCE data, most *B. v. magellanicus* in the mitochondrial trees form their own clade divergent from all other subspecies. This clade is 5.95% divergent from the northern clade using the mitochondrial

alignment that included all genes. However, *B. v. magellanicus* MSB 35949 falls within the northern clade and is one of three northernmost *B. v. magellanicus* samples.

## Gene flow

Significant F-branch statistics between *B. v. nigrescens* (LSUMZ 30041) from Ecuador and Peruvian *B. v. magellanicus* support Nuclear gene flow between *B. v. magellanicus* and non-*magellanicus B. virginianus* populations in the Andes (Fig. S4). We do not have the sampling to assess gene flow between *B. v. magellanicus* and other *B. virginianus* subspecies at the eastern limit of the *B. v. magellanicus* range. We also identified consistent gene flow between *B. v. nigrescens* (FMNH 102981) and much of the northern clade (Fig. S4). Other areas of the tree also showed evidence of gene flow, but gene flow is expected at the subspecific level (*Price, 2008*).

## Audio recording sonograms

Sonograms of the six audio recordings from Xeno-Canto illustrate the vocal differences between populations of *B. v. magellanicus* and northern *B. virginianus* described by *López-Lanús (2015)* (Fig. 4). The two recordings from *B. v. magellanicus* had two notes followed by approximately a two second trill. The song from the four northern *B. virginianus* recordings had three to five notes. These songs are clearly distinguishable without formal vocal analyses due to the strong differentiation in amount and length of notes.

## DISCUSSION

The analyses in this article address genetic variation and differentiation across the broad geographic range of *B. virginianus.* Our results confirm a deep genetic divergence between populations in the central and southern Andes (*B. v. magellanicus*) and all other samples (12 sampled subspecies). The Marañón Valley is a common biogeographic barrier for higher-elevation birds in South America due to a steep transition between wet mountain regions and a dry valley (*Winger & Bates, 2015*) and may limit the distributions of these two taxa. Although this valley is a relatively abrupt divide for many high elevation species, we lack dense sampling of *B. virginianus* close to this valley to accurately assess more fine-scale patterns across this divide. In the current dataset, we also lack extensive geographic sampling in South America more broadly, which precludes addressing outstanding questions of geographic structure within South America, particularly of isolated populations far east of the Andes. The mean mitochondrial distance between *B. v. magellanicus* and northern *B. virginianus* populations (mean between-group corrected Jukes-Cantor divergence = 5.95% using all mitochondrial genes) is well over the 2% divergence that has been suggested as indicative of species-level divergence in birds (*Price, 2008*). A clear phylogenetic split is present in the nuclear UCE data as well. In addition to the data presented here, previous authors have noted vocal, body size, and plumage differences between northern and southern populations (*López-Lanús, 2015*; *König, Heidrich & Wink, 1996*; *Traylor, 1958*). *B. v. magellanicus* is a smaller owl with finer barring and comparatively small bill and feet in relation to overall body size (*König, Heidrich & Wink, 1996*). The morphological variation between these two species may have a more abrupt transition than between other

*B. virginianus* populations, but sufficiently dense sampling from this region is not available to understand the transition without further work.

Despite complex patterns of morphological variation across the entire distribution of *B. virginianus*, vocal variation is minimal within the northern and southern groups, but starkly different between them (*López-Lanús, 2015*). The song of northern populations of *B. virginianus* has three to five longer notes, whereas the song of *B. v. magellanicus* has two longer notes with a trill at the end (Fig. 4). Previous vocal work has suggested that *B. v. nigrescens* should also be considered a separate species due to their shorter song (see Fig. 4D), but our phylogenetic analyses do not identify this subspecies as monophyletic in either nuclear or mitochondrial datasets (*López-Lanús, 2015*).

Audio recordings on Xeno-Canto suggest that *B. v. magellanicus* does not extend east into the South American lowlands. Audio recordings of individuals that sing a song typical of *B. v. magellanicus* include recordings from Cajamarca Region, Peru (XC139660), Cordoba, Argentina (XC51622), and Salta, Argentina (XC212734), whereas recordings of individuals giving typical northern songs (*i.e.,* non-*magellanicus*) occur within southern South America as far west as Buenos Aires, Argentina (XC645511) and Beni, Bolivia (XC149520). The non-overlap of *B. v. magellanicus* and other *B. virginianus* subspecies may be due to habitat partitioning, in which non-*magellanicus B. virginianus* subspecies may be more forest-associated (*Roesler, 2022*).

We found one instance of incongruence between the topologies of the UCE and mitochondrial trees regarding the deep divergence between northern and central Andes. Although the UCE data showed a clear divide between populations, the mitochondrial data showed that one individual from Lima, Peru (subspecies *magellanicus* by range and nuclear data) had the mitochondrial genome of the non-*magellanicus* group. This mitonuclear discordance is most commonly produced by either incomplete lineage sorting (*i.e.,* deep coalescence) or gene flow between populations (*Maddison, 1997*). F-branch analyses assessed these two options (incomplete lineage sorting *versus* gene flow) using the nuclear sequence data and identified significant gene flow between *B. v. nigrescens* in Ecuador and *B. v. magellanicus* in Peru. The mitochondrial and UCE data were bioinformatically extracted from the same DNA sequencing reaction, so the mitonuclear discordance should not be caused by contamination. Contamination from other samples during DNA extraction or library preparation is possible, but unlikely given the consistently southern nuclear data recovered from the individual with the northern mitochondrial sequence. The combination of mitonuclear discordance and significant signal of nuclear gene flow between *B. v. nigrescens* in Ecuador and *B. v. magellanicus* in Peru suggests some level of ongoing or historical gene flow between these two distinctive populations. The nature and extent of this mitonuclear discordance and the gene flow in northern Peru can only be determined with denser sampling.

Our results also confirm the shallow structure seen in the southwestern United States (*Dickerman, Mcnew & Witt, 2013*) and show that this shallow structure extends across all populations outside of the central and southern Andes. Northern South American populations clearly belong within the northern *B. virginianus* clade but display shallow population structure. Individuals sampled from across North America in particular

have low genetic divergence and little support for phylogenetic relationships, indicating prevailing gene flow, range expansion, or recent divergence. *Bubo virginianus* are known to be philopatric, but mixed evidence for yearly migration or dispersal has been documented (*Baumgartner, 1939*; *Dickerman, Mcnew & Witt, 2013*; *Houston, 1999*). Although short distance migration is known in some subspecies (*Houston, 1978*; *Dickerman, Mcnew & Witt, 2013*), these movements should not have a large effect on the interpretation of our data. We used specimens collected during breeding season where available, but acknowledge that wintering birds might be present when southern birds start breeding. The lack of population structure in North America alleviates major concerns about wintering birds causing misleading geographic patterns of genetic structure. More detailed geographic sampling and data types more suited to detecting subtle population structure are needed to assess any genetic patterns among the numerous subspecies recognized in North America.

## CONCLUSIONS

In this article we present the first genome-scale, range-wide molecular analysis of genetic variation in *B. virginianus*. We found little genetic differentiation among populations throughout North America, Central America, and northern South America north of the central Andes. In contrast to that shallow differentiatioin, we identified substantial divergence, in both nuclear and mitochondrial DNA sequences, between populations of the central and southern Andes and all others. These southern populations, currently assigned to the subspecies *B. v. magellanicus,* show morphological, vocal, and genetic differences consistent with species-level differentiation. Despite deep mitochondrial and nuclear divergence between *B. v. magellanicus* and other *B. virginianus*, we found one case of mitonuclear discordance and we observe evidence of nuclear gene flow across the north-south break in northern Peru. Although our data do not identify the geographic distribution of the transition area between these putative species, the vocal and genetic data presented here provide support for species-level differentiation. Additional genetic and vocal data are needed to assess the extent of interactions between the two populations and yield a more thorough understanding of the dynamics between these two putative species.

## ACKNOWLEDGEMENTS

We would like to thank the many museums that loaned tissues for this project: Academy of Natural Sciences of Drexel University, California Academy of Sciences, Carnegie Museum of Natural History, Denver Museum of Nature & Science, Field Museum of Natural History, University of Kansas Biodiversity Institute, Natural History Museum of Los Angeles County, Louisiana State University Museum of Natural Science, Museum of Comparative Zoology, Harvard University, Museum of Southwestern Biology, Royal Ontario Museum, University of Alaska Museum, University of Michigan Museum of Zoology, National Museum of Natural History, University of Washington Burke Museum. This manuscript was greatly improved by comments from three anonymous reviewers and editing from A. Townsend Peterson, Rafe M. Brown, and Jennifer A. Raff. Mark B. Robbins gave valuable advice while designing the project.

### Funding

The funding for this project was supported by the Panorama Grant from the KU Biodiversity Institute and the University of Kansas Genome Sequencing Core Voucher. Emily N. Ostrow was supported by the National Science Foundation Graduate Research Fellowship Program. There was no additional external funding received for this study. The funders had no role in study design, data collection and analysis, decision to publish, or preparation of the manuscript.

### Grant Disclosures

The following grant information was disclosed by the authors:
Panorama Grant from the KU Biodiversity Institute and the University of Kansas Genome Sequencing Core Voucher.
National Science Foundation Graduate Research Fellowship Program.

### Competing Interests

The authors declare there are no competing interests.

### Author Contributions

- Emily N. Ostrow conceived and designed the experiments, performed the experiments, analyzed the data, prepared figures and/or tables, authored or reviewed drafts of the article, and approved the final draft.
- Lucas H. DeCicco performed the experiments, authored or reviewed drafts of the article, and approved the final draft.
- Robert G. Moyle conceived and designed the experiments, authored or reviewed drafts of the article, and approved the final draft.

### Data Availability

The sequencing data generated for this study is available on NCBI: PRJNA879795.
The original code is available at Zenodo: Emily N. Ostrow. (2023). emilyostrow/ReplaceUninformativeSites: ReplaceUninformativeSites (for Publication). Zenodo. https://doi.org/10.5281/zenodo.8084429.

### Supplemental Information

Supplemental information for this article can be found online at http://dx.doi.org/10.7717/peerj.15787#supplemental-information.

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
