# Peer review of "Range-wide phylogenomics of the Great Horned Owl (Bubo virginianus) reveals deep north-south divergence in northern Peru"

_PeerJ, doi:10.7717/peerj.15787_

## Round 0.1 · original submission · Minor Revisions

Two reviewers called for minor revisions and one for major revisions. Please address all the points raised by the reviewers, with special focus on reviewer #3's comments.

Reviewer 1 ·

Basic reporting

In the manuscript titled “Range-wide phylogenomics of Great Horned Owls (Bubo virginianus) reveals deep North-South divergence across the Marañón Valley, Peru”, the authors investigated the phylogenetic relationship among individuals from a single owl species, Bubo virginianus, using sequences of mitochondria and ultra-conserved elements (UCEs) of twenty-seven samples across the range of the species. Recently a subspecies of B. virginianus, Bubo virginianus magellanicus, was recognized as a separate species by the International Ornithological Congress. The authors confirmed that B. v. magellanicus is divergent enough and should be treated as an independent species, using 1898 SNPs for UCEs (around 4200 UCE loci) and 11,347 bp of sequences from mtDNA.

Experimental design

no comment

Validity of the findings

The main contribution of this manuscript is offering genomic data for B. v. magellanicus, for which very limited genetic data (2 individuals) were previously available. Although I do appreciate it, I think it requires minor revision before ready for publication.

Additional comments

Major comments:
・For phylogenetic analyses, the authors used Maximum-likelihood and quartet-based methods. However, both are Maximum-likelihood methods. Why don’t you use a distance-based method, such as the Neighbor-joining method? The NJ method is implemented in MEGA software and is very fast to compute. I guess the authors tried to remove the effects of incomplete lineage sorting, but it would still be nice and informative to see NJ trees as basic default information and to confirm the consistency with the ML trees. Please include NJ trees in the manuscript.

・Lines 232-234. “Within the northern clade, relationships generally followed latitude, with earlier-diverging lineages belonging to more southern taxa.” and Line 242. “loosely following latitude”
I found this description is not very accurate. Rather it’s better to point out that B. v. nigrescens and B. v. nacurutu (southern taxa) are at the basal position of the trees (both for UCEs and mt genomes) though B. v. nigrescens does not form a monophyletic clade. The numbers of samples and SNPs are limited, but the basal positions of the southern taxa may suggest the southern origin of this species. The inner clade does not seem to follow latitude.

・Lines 301-303. XC139660, XC51622, and XC212734 don’t appear in the legend of Figure 4. Why is it?

・In the legend of Figure 4, there is an explanation about Figure 4(G), which does not exist.

・The sampling location of the audio recordings is not very clear to me. Please include a geographical map and a sample list for them.


Minor comments:
・Line 267. “Marinon Valley” is a typo of “Marañón Valley”?

Annotated reviews are not available for download in order to protect the identity of reviewers who chose to remain anonymous.

Reviewer 2 ·

Basic reporting

This is a very well written article that provides a compelling argument. The referenced literature are all relevant and I see no issues with formatting. The figures are clear and easy to interpret.

Experimental design

The experimental design is sound building a case using information gleaned from across other areas of the focal organism's range. Using genetic, phenotypic and acoustic information creates a nice comparison in which to highlight differences relative to the geographic area of focus. The phylogentic trees provided in the figures supplement the findings given in text and other figures. The genetic approaches applied all seem sound and appropriate, though I admit they are outside my area of expertise.

Validity of the findings

The findings are nicely presented using clear logic. Outside the area of this article but it would be interesting to see a map, perhaps in the supplemental material, of where Bubo virginianus magellanicus has been observed versus B. v. nacurutu and B. v. nigrescens. Do these groups overlap in their range? Are there elevation, or other abiotic differences that separate these groups? Figure 1 shows a nice separation among groups and their distributions but it is less clear why these differences occur.

Additional comments

Thank you for presenting a nice, clear paper. I found it nice to read.

Reviewer 3 ·

Basic reporting

This study is the first to attempt range-wide genetic and phylogenetic structure and the Great Horned Owls of North and South America. The background provided in the introduction sets up the study and questions nicely. The methods chosen are very appropriate. Excellent use is made of museum specimens from a variety of sources, and it would be difficult to complete the sampling for a project like this without museum collections. The authors use a combination of toepads and tissues, which allowed them to expand their sampling considerably.

The molecular markers chosen were UCE’s and mt-genomes, both appropriate for the questions being asked. The lab and analytical methods chosen were valid and sufficient.

An analysis of vocalizations was added in order to address the ‘species-limits’ question. However, this felt a bit tacked on because it depended on only six high-quality recordings, from disparate parts of the species range. I don’t think six was sufficient to be able to account for individual and sex variation in statistical models. The present paper writes on lines 282-3 that previous work has noted vocal differences between northern and southern GHOW’s, but the present vocal work doesn’t seem to go beyond ‘noting’ differences. Available sampling is simply too sparse to pinpoint where the vocalization type turns over, except that it seems to occur between Azuay and Cajamarca; but the lack of formal analysis in this paper means that it doesn’t clearly add anything to information that is already easily available in the public sphere.

The main interest of the authors seems to be in making an argument for species status of magellanicus, the most divergent of the 15 GHOW subspecies. I found the argument somewhat convincing – nearly 6% mtDNA divergence, UCE monophyly, distinct vocalizations. However, the case also has some striking weaknesses — extremely sparse sampling in both genetic and vocal data across the boundaries between magellanicus and adjacent subspecies. This leaves a substantial amount of uncertainty about species limits, range limits, reproductive inter-compatibility, and gene flow. With only three samples from the western Central Andes of Peru, one of them (33%) had mismatched mtDNA! That suggests that the entire central Peruvian population is polymorphic for the two most divergent mtDNA clades in GHOW – a neat finding and one that certainly raises the need for more sampling and more analyses. One obvious thing to do would be to use the UCE data at hand to test for evidence of gene flow from Ecuador into central Peru, perhaps using TreeMix (or perhaps any of numerous other available methods). I would suggest doing so as part of a revision of this manuscript; although the study is still valid without it, the assertions about species status are unnecessarily weak without it.

I would suggest reducing the emphasis on the Marañon Valley in this paper, and instead focusing the paper on the Northern Andes vs. Central Andes (the two zoogeographic regions that are divided by the North Peruvian Low and the Marañon Gap). I would also suggest going to great lengths to emphasize that we do not know where the two taxa turn over, nor do we know the clines widths or centers for plumage, vocalizations, size, mtDNA, or UCE’s – but the out of place mtDNA gives us evidence that those clines are not all in the same place. Regarding the possibility that ILS caused the mismatched mtDNA: sure, it seems smart to list it as a possible cause of mitonuclear discordance… but in this case it seems far-fetched because of the deep divergence and the fact that the mismatched individual is closer to the contact zone than most other samples in the study. The mismatch is a strong indicator of gene flow from the Northern Andes to the Central Andes. It will be important, in that respect, to test for geneflow using UCE’s and further hone our understanding of connectivity across this divide, the deepest one in GHOW’s.

Line 254: “However, B. v. magellanicus MSB 35949 falls within the northern clade and is one of three northernmost B. v. magellanicus samples, relatively close to the Marañón Valley.” I disagree that this is close to the Marañon Valley – that bird was collected in Lima, Peru, quite far from the Marañon.

Line 261: 3 or 4 notes? Later it says 4 or 5 notes (clearly they sometimes have 5… do some also have only 3? Clarify this).

Line 271-2: The Marañon is not a barrier for high altitude species per se, just those that are restricted to humid-forest habitats. It’s not at all obvious why it would create a dispersal barrier for GHOW given that they should be in the western cordillera and able to tolerate lower and/or drier conditions.

Line 288: “Despite complex patterns of morphological variation across the entire distribution of B. virginianus, vocal variation is minimal within the northern and southern groups, but starkly different between them.” There is not enough data in this paper to support this statement.

Museum names should be written out in acknowledgments in my opinion. The acronyms are known to relatively few people (mainly curators and collection managers!), and who knows how long vertnet will be around. They should also be written out in the caption to Table 1. And I would suggest that the acronyms be added to Figure 3, where there are only numbers with no museum ID’s (not a best practice).

Fig. 2: why aren’t the outgroup taxa in the fresh tissue analysis? That would be nice to see.

Experimental design

na

Validity of the findings

na

Additional comments

na

---

## Round 0.2 · accepted · Accept

The authors have addressed all reviewer comments. The manuscript is ready for publication.